# Urinary Proteome Changes during Pregnancy in Rats

**DOI:** 10.3390/biom13010034

**Published:** 2022-12-24

**Authors:** Shuxuan Tang, Youhe Gao

**Affiliations:** Gene Engineering Drug and Biotechnology Beijing Key Laboratory, College of Life Sciences, Beijing Normal University, Beijing 100875, China

**Keywords:** urinary proteome, pregnancy, embryonic development, coagulation pathway

## Abstract

Pregnancy involves a significant number of physiological changes. A normal pregnancy is essential to ensure healthy maternal and fetal development. We sought to explore whether the urinary proteome could reflect the pregnancy process. Urine samples were collected from pregnant and control rats on various gestational days. The urinary proteome was profiled by liquid chromatography coupled with tandem mass spectrometry (LC–MS/MS), and differential proteins were obtained by comparing to the gestational day 1 of the same group at each time point. Many pathways related to embryo implantation and trophoblast differentiation were enriched in the early days in urine. Liver, kidney, and bone development started early to be enriched in the pregnant group, but not in the control group. Interestingly, the developmental processes of the fetal heart such as heart looping and endocardial cushion formation could be seen in urine of pregnant rats. Moreover, the timings were consistent with those of embryological studies. The timing of the surfactant appearance in urine was right before birth. The differential proteins related to pancreas development appeared in urine at the time during reported time of pancreatic cell proliferation and differentiation. These processes were enriched only in the pregnant group and not in the control group. Furthermore, coagulation-associated pathways were found to be increasingly prominent before labor. Our results indicated that the urine proteome of pregnant rats can reflect the process of pregnancy, even fetal embryonic development. Maternal urinary proteome detection was earlier than the developmental time point of tissue sections observed by microscopy.

## 1. Introduction

Pregnancy is characterized by dynamic physiological changes in the female reproductive system [1]. Fetal development surveillance ensures normal embryonic development and also aids in the detection of fetal defects.

As a filtrate of the blood, urine bears no need or mechanism to be stable. Urine is the place where most of the wastes in blood are dumped into, and thus it tolerates changes to a much higher degree. Biomarkers are the measurable changes associated with a physiological or pathophysiological process. Therefore, they are more likely to be magnified and detectable in urine than their counterparts in blood [2]. The urinary proteome reflects changes in different types of diseases, such as Parkinson’s disease (PD) [3], autism [4], glioma [5], liver cancer [6], bone cancer metastasis [7], colitis [8], and myocarditis [9]. Animal experiments are characterized by a short gestational period and are less affected by certain factors, such as living habits, better ensuring the consistency of individuals. Thus, we intended to explore if urine in pregnant rats can reflect pregnancy and embryonic fetal development processes.

## 2. Materials and Methods

### 2.1. Experimental Animals and Groups

Pregnant Wistar rats (gestational day 1 d, n = 15, 190 ± 40 g) and control Wistar rats (n = 5, 200 ± 40 g) were purchased from Beijing Vital River Laboratory Animal Technology Co., Ltd. The day on which the vaginal plug of the female rat was examined was gestational day 1 d (GD 1 d). The animal room was maintained at a temperature of 25 ± 2 °C, a humidity of 65–70%, and a 12 h light–dark cycle. This study was conducted in accordance with guidelines developed by the Institutional Animal Care and Use Committee of Peking Union Medical College. This experiment was approved by the Ethics and Animal Welfare Committee of Beijing Normal University (CLS-EAW-2020-022).

In this study, 8–10-week-old Wistar rats were divided into two groups: the pregnant group and control group. The urine samples of the two groups were collected in two batches using rat metabolic cages for 6–8 h. The first batch of urine samples was collected from pregnant rats (n = 10): gestational days 1, 4, 7, 11, 14, 16, 18, 20 d. The second batch of urine samples were collected from pregnant rats (n = 5) and control rats (n = 5): pregnant group gestational days 1, 4, 7, 11 d; control group 1 d, 4 d, 7 d, 11 d, 14 d, 16 d, 18 d, 20 d.

### 2.2. Urine Sample Preparation

Random urine samples during the day were collected for 6–8 h and then stored at −80 °C for later sample processing. Urine (4–6 mL) for each sample was centrifuged at 14,000× *g* for 30 min at 4 °C. The supernatant was added to at least 3 volumes of ethanol. The solution was placed at −20 °C overnight for precipitation. Then, 100 µL of lysis buffer (8 mol/L urea, 2 mol/L thiourea, 50 mmol/L Tris, and 25 mmol/L dithiothreitol) was added to obtain all urine proteins. Protein concentrations were quantified by the Bradford kit assay.

Each sample was digested with 100 µg of protein using the FASP method [10]. The sample was loaded onto a 10 kD ultracentrifugation filter and washed twice with UA solution (8 mol/L urea, 0.1 mol/L Tris-HCl; pH 8.5), and the protein sample was reduced with 20 mmol/L dithiothreitol (37 °C) for 1 h and then alkylated with 50 mmol/L iodoacetamide (IAA, Sigma) for 45 min in the dark. Next, the samples were washed twice with UA and three times with 25 mM NH_4_HCO_3_ and then digested overnight at 37 °C with trypsin (ratio of enzyme:protein 1:50). The digested peptides were eluted from 10 kD filters and desalted using HLB (Waters, Milford, MA, USA).

### 2.3. Peptide Fractionation

The peptides were diluted to 0.0.5 µg/µL with 1%_0_ formic acid water, and all samples (1 µg/sample) were mixed into QC samples. The pooled sample was loaded on high-pH reversed-phase spin columns (Thermo Scientific USA, 84868). Eluents with different concentrations of acetonitrile (5, 7.5, 10, 12.5, 15, 17.5, 20, and 50%) were added to reversed-phase fractionation spin columns, and the peptides were eluted with different concentrations of acetonitrile to obtain 8 different gradients of fractions. These peptides were lyophilized in vacuo and reconstituted in 20 µL of 1% formic acid in water.

### 2.4. LC–MS/MS Analysis

To correct the retention time of the extracted peaks, iRT was added to each sample according to the ratio of peptides: iRT volume ratio of 10:1. One microgram of each sample was loaded at a rate of 300 µL/min through a trap column (75 μm × 2 cm, 3 μm, C18, 100 Å). Peptides were eluted with a gradient of 4–35% buffer B (0.1% formic acid in 80% acetonitrile, flow rate at 0.3 µL/min) for 90 min.

Eight fractions were analyzed with a mass spectrometer in DDA mode. The parameters of MS in DDA mode were set as follows: the full MS scan was acquired from *m/z* 350–1550 with an orbitrap resolution of 12,000. The MS/MS scan was acquired in orbitrap mode with a resolution of 30,000. The HCD collision energy was set to 30%. The AGC target was set to 5 × 10^4^, and the maximum injection time was 45 ms. Then, the mass spectrometer was changed to the DIA mode, and the individual samples were analyzed in DIA-MS mode. The variable isolation window of the DIA method with 36 windows was adopted. The full scan was set at a resolution of 60,000 over a range of *m/z* 350–15,000. All samples were mixed to obtain QC for DIA mode analysis.

### 2.5. Data Processing

The ten raw files were searched against the Swiss-Prot rat database with SEQUEST HT by Proteome Discoverer (version 2.1; Thermo Scientific). A maximum of two missed cleavage sites were allowed in the trypsin digestion. The parent ion mass tolerances were set to 10 ppm, and fragment ion mass tolerance was set to 0.02 Da. The cutoff at the precursor and protein levels was less than 1%.

The QC and individual files were imported into Spectronaut Pulsar X. Protein inference was performed using the implemented IDPicker algorithm [11]. The Q value (FDR) was set at less than 1% for proteins and precursors. Quantitative results were processed by Spectronaut. QC samples were used to evaluate the stability of the mass spectrometry. The sequential-KNN method was used to fill in the missing values of QC at website https://www.omicsolution.org/wkomics/main (accessed on 20 December) Proteins with a coefficient of variation of QC < 0.3 were selected for subsequent screening.

### 2.6. Statistical Analysis

Comparison of data at different time points was analyzed with a two-sided *t*-test. Differential proteins in the pregnant group were identified by comparing the urinary proteome at each time point with GD 1 d (the first day of fertilization). Similarly, the differential proteins of the control group were identified for each time point compared with the urinary proteome of control 0 d. Differential protein screening criteria: at least two specific peptides in the protein were identified, fold change ≥1.5 or ≤0.67, *p* < 0.05 calculated by a two-sided *t*-test. The average number of protein groups per sample identified by each batch mass spectrometry, are presented as the mean ± standard deviation (SD).

### 2.7. Bioinformatics and Functional Analysis

Biological processes (BPs) were obtained by GO analysis of the differential proteins using the DAVID website. Biological pathway analysis and disease/biological function analysis were performed with differential proteins by IPA software (Ingenuity Systems, Mountain View, CA, USA). Visual mapping and data preprocessing were conducted on the WuKong platform (https://www.omicsolution.org/wkomics/main/ accessed on 20 December). The graph was generated by graphpad Prism 8 to compare the expression level changes of related proteins involved in the blood coagulation process, which was enriched by the IPA biological pathway, and the number of samples at each pregnant time point was n = 3 when inputting parameters.

## 3. Results and Discussion

### 3.1. Urinary Proteome Changes and Function Analysis in Pregnant Rats and the Control Group

In this study, urine samples from the rats in the pregnant group (n = 15) and the control group (n = 5) were collected at GD 1 d, GD 4 d, GD 7 d, GD 11 d, GD 14 d, GD 16 d, GD 18 d, and GD 20 d and 0 d, 4 d, 7 d, 11 d, 14 d, 16 d, 18 d, and 20 d. The number of urine proteins identified in the two groups is shown in Table 1. A total of 2014 proteins were identified in the first batch, with an average of 1703 protein groups identified per sample. A total of 1868 proteins were identified in the second batch, with an average of 1498 protein groups identified per sample (Table 1). Protein identification was based on at least two unique peptides, and the missing values of the mix were filled through the Wukong platform. Finally, the CV value of QC was evaluated, and the protein with CV < 0.3 was selected. Proteins remaining as the final dataset from the first batch and the second batch were 1199 and 1289 for further analysis, respectively (Table 1).

The differential proteins of the pregnancy group at each time point were obtained by comparison with the urinary proteome of the same batch at GD 1 d. The differential proteins of the control group were obtained by comparison with the urinary proteome at 0 d at each time point. A Venn diagram of differential proteins in the control and pregnancy groups is presented in Figure 1A. There were 83 proteins in common between the control and pregnancy groups and 423 unique proteins in the pregnancy group, which were much more than those in the control group. We aimed to explore the physiological changes related to pregnancy from the urinary proteome, and the differential proteins of the pregnancy and control groups were enriched for biological processes and IPA biological classical pathways (Appendix A), respectively. The differential proteins of the pregnancy and control groups were enriched for biological processes and IPA biological classical pathways, respectively. The Venn diagrams of the enriched results are shown in Figure 1B,C. On the basis of a comparison of the number of differential proteins (Figure 1A), biological processes (Figure 1B), and IPA biological classical pathways (Figure 1C) between the control and pregnant groups, the number in the pregnant group was found to be much larger than that in the control group, suggesting that the changes in pregnancy reflected in urine were much greater than the changes associated with the growth and development of the rats themselves.

### 3.2. Urine Proteome Changes during Pregnancy

The differential proteins at each time point were identified and compared with those on GD 1 d. The urinary proteome changes on the first day after fertilization were reflected by the comparison of the urinary proteome at GD 1 d and the control group at 0 d. After screening (FC ≥ 1.5 or ≤0.67, *p* < 0.05), 64, 56, 39, 69, 52, 69, 220, and 200 differential proteins were obtained at these time points in the pregnant group. A Venn diagram of the differential proteins in the pregnant group at each time point is shown in Figure 2. The largest number of differential proteins were identified on GD 18 d and GD 20 d, which indicated that pregnancy changed greatly during the period of pregnancy.

### 3.3. Functional Analysis of the Differential Proteins in the Pregnant Group—Implantation Period

The implantation period of the rats started on GD 5 d and was completed on GD 7 d. The biological processes of urinary proteome enrichment of pregnant rats on GD 1 d, GD 4 d, and GD 7 d are shown in Figure 3 and Appendix A. Biological processes related to growth and development were enriched on the first day after fertilization (GD 1 d), including liver development, kidney development, and bone development. Liver and kidney development were earlier than in embryological studies [12,13] (Figure 3). Other biological pathways are the interleukin-17-mediated signaling pathway and response to L-ascorbic acid. Interleukin-17 activates the PPAR-γ/RXR-α signaling pathway and promotes the proliferation, migration, and invasion of trophoblast cells [14]. L-ascorbic acid (VitC) inhibits placental trophoblast apoptosis and increases syncytiotrophoblast differentiation, and VitC is also involved in the synthesis of placental steroids and polypeptide hormones [15].

The development and growth-related biological pathways displayed by differential proteins on GD 4 d in the pregnant group include liver development and glomerular visceral epithelial cell development. Pathways affecting implantation include positive regulation of insulin secretion involved in the cellular response to glucose stimulus (Figure 3). Insulin can directly impair ovarian function during embryo implantation and insulin-induced ovarian autophagy imbalance.

Many biological pathways and processes were related to implantation on GD 7 d, such as the L-ascorbic acid metabolic process, L-ascorbic acid biosynthetic process, and tryptophan degradation to 2-amino-3-carboxymuconate semialdehyde. Large amounts of ascorbic acid exert anti-inflammatory and immunological effects that are beneficial for embryo implantation [16]. Tryptophan is a key regulator of embryogenesis and embryo implantation during pregnancy. High L-Trp levels (500 and 1000 mM) induce cell cycle arrest in the G1 phase and inhibit cell proliferation in porcine trophectoderm cells [17].

Changes in urinary proteome exhibited during the implantation period of rats were mainly related to growth and development, implantation, and placental growth, and the implantation-related pathways were only enriched in this period and did not appear in other periods during the process of embryo implantation.

### 3.4. Functional Analysis of the Differential Proteins in the Pregnant Group—Embryonic Organ Development

Embryonic organ development is an important event in the process of embryo maturation. Rat embryonic organs begin to develop in the middle and late stages of pregnancy. The processes and pathways enriched by changes in the urinary proteome were analyzed to explore the growth and development of embryos.

Previous studies have shown that rat embryos begin to form a cardiac septum on GD 11 d and form a mature s-loop; on GD 14 d, the dorsal and ventral atrioventricular cushions begin to fuse, and the atrioventricular septum begins to appear; on GD 16 d, the heart matures, and the interventricular orifice is completely closed. The developmental period of the rat heart was reported to be from GD 9 d to GD 16 d [18]. The development of the heart was reflected in the urine, as shown in Figure 4. The study showed the biological process of heart looping on GD 11 d, the process of positive regulation of epithelial to mesenchymal transition involved in endocardial cushion formation on GD 14 d, and the cellular response to platelet-derived growth factor stimulus (PDGF) on GD 16 d. Clearly, the biological processes that emerged at GD 11 d and GD 14 d were completely consistent with the previously reported time course of cardiac development. PDGF-BB increased myocardium and myofibril compaction, and PDGF-AA increased myocardial fiber differentiation [19]. In addition, angiogenesis involved in coronary vascular morphogenesis and heart development pathways was associated with cardiac coronary development.

Previous studies have shown that the pancreas of the rat developed a pancreatic bud at GD 9.5 d; the mesenchyme pushed the bulge around the dorsal gut to initiate pancreatic formation at GD 11 d; the mesenchyme surrounded the dorsal gut, pushing the bulge to initiate the formation of the pancreas at GD11 d–GD 11.5 d; and GD 14 d–GD 18.5 d is a critical period for rat pancreatic cell proliferation, differentiation, and structure formation [20,21]. The present study shows that the maternal urinary proteome reflected the process of pancreatic B cell proliferation at GD 14 d (Figure 5), which is consistent with the period of pancreatic cell proliferation [21].

The development of embryonic lungs is closely related to the survival rate of the fetus after birth. The development of the rat embryonic lung begins on GD 11 d, and the lung primordium appears on GD 11 d; airway and pleura form on GD 13 d; bronchial development begins on GD 14 d; pulmonary alveoli and pulmonary vasculature generate between GD 14 d and 16 d; and alveolar and alveolar surfactant formation occur between GD 18.5 d and GD 20 d [22]. The results of this study showed that branching morphogenesis of an epithelial tube appeared on GD 11 d; the epidermal growth factor receptor signaling pathway (EGF) appeared on GD 16 d; and glucocorticoid receptor signaling, the Wnt/β-catenin pathway, and PCP (planar cell polarity) appeared on GD 18 d (Figure 6). The pathways emerging on GD 18 d are associated with alveolar surfactant formation, such as glucocorticoid receptor signaling, which increases dramatically in fetal plasma in late pregnancy, stimulating fetal pulmonary surfactant synthesis [23]. Wnt/β-catenin is associated with airway smooth muscle cell proliferation and alveolar development [24]. Alveolar formation during this period is approximately the same as previously reported. PCP is involved in the directional movement of cilia in the lungs [25]. EGF at GD 16 d is an important regulator of cell differentiation and fetal pulmonary surfactant synthesis [26], possibly in preparation for the next stage of alveolar synthesis.

These pathways that corresponded to the timing of organ development only appeared in the pregnant group and were not found in the control group (Appendix A), indicating that these pathways originated from embryonic development rather than the rat’s own growth and development. In addition, the process of growth and development of other organs over time is shown in Figure 7. The pathways related to kidney development were glomerular visceral epithelial cell development and glomerular filtration. These two pathways were most prominent on GD 20 d, and glomerular visceral epithelial cell development was enriched on GD 4 d, GD 11 d, GD 16 d, and GD 20 d; glomerular filtration was enriched on GD 16 d and GD 20 d; and liver development was enriched on GD 0 d, GD 4 d, and GD 11 d. Hair follicle morphogenesis was enriched on GD 11 d; salivary gland cavitation was enriched on GD 16 d; and skeletal muscle tissue development and neural crest cell migration were enriched on GD 18 d (Appendix A). These results suggest that the urine in pregnant rats may reflect the status of embryonic growth and has the potential to be an effective strategy for monitoring normal embryonic development.

### 3.5. Functional Analysis of the Differential Proteins in the Pregnant Group Related to the Maternal Coagulation System

From the first to the third trimester, with increasing gestational age, coagulation function is gradually enhanced, including increased thrombin generation and fibrinolysis. These two factors co-occur and, in most cases, can lead to a hypercoagulable state. These changes are thought to maintain placental function during pregnancy and prevent massive blood loss during delivery [27].

Figure 8A showed the biological pathways on GD 20 d, with the extrinsic prothrombin activation pathway, GP6 signaling pathway, coagulation system, intrinsic prothrombin activation pathway, and prostanoid biosynthesis being related to coagulation function (Figure 8A). The interaction of platelets with collagen via the GP6 receptor leads to platelet activation and adhesion, a process required for thrombosis. GP6 is a platelet transmembrane glycoprotein that plays an important role in collagen-initiated signal transduction and platelet procoagulant activity [28]. Prostanoid biosynthesis and endogenous prostaglandin production are associated with labor in humans, prostaglandin levels continue to rise during labor, and prostaglandins appear to be involved in the increased contractile activity observed during labor [29]. The changes in these biological processes related to procoagulation are shown in Figure 8B. The trend of coagulation biological process between GD 14 d and GD 20 d showed that the pathway was significantly unregulated on GD 20 d. In addition, compared with GD 14 d, the number of coagulation pathways also increased significantly on GD 20 d, suggesting that the coagulation function of pregnant rat continued to increase before delivery.

To assess changes in coagulation-related proteins, their relative abundances were analyzed during pregnancy. MOR8A3 glycoprotein 6 (platelet) was expressed at low levels in early pregnancy but upregulated at GD 14 d and maintained at high levels before delivery (GD 20 d) (Figure 9A). Q7TQ70 fibrinogen alpha chain was an important component in the formation of insoluble fibrin matrix (Figure 9B). Fibrin was one of the main components of blood clots and played an important role in hemostasis, also being upregulated at GD 14 d and remaining higher than the first trimester until delivery. Similarly, fold change of the procoagulant-related proteins became larger over time (Figure 9C). Changes in the coagulation system revealed by the urinary proteome were consistent with reported changes before labor.

Although this study is based on animal experiments, the results suggest that more clinical samples need to be included to explore the pregnancy process reflected by the human urine proteome in the future. The normal urine proteome of pregnant women may serve as a clinical database, laying a foundation for further clinical research. The normal urine proteome of pregnant women may serve as a clinical database for the obstetric detection of abnormal pregnancy.

## 4. Conclusions

Our results show that the urinary proteome can reflect changes in different gestational stages. During the period from fertilization to implantation in rats, many pathways related to embryo implantation and trophoblast differentiation were enriched on GD 1 d, GD 4 d, and GD 7 d. Liver, kidney, and bone development started early to be enriched at this period. During the second and third trimesters, pathways related to the development of embryonic organs, such as the development of the heart, heart looping, and endocardial cushion formation, were consistent with the developmental times reported in embryological studies. The timing of the emergence of surfactant synthesis signaling pathways is in good agreement with embryological studies. The development of pancreatic islet B cells also occurs during the main proliferation and differentiation periods of pancreatic cells. On day 20 of gestation, many coagulation-related pathways were enriched, and the significance of coagulation-related proteins and pathways continued to increase, which was consistent with the change trend of prenatal coagulation function reported study.

In summary, our results indicated that the urine proteome of pregnant rats can reflect the process of pregnancy, even fetal embryonic development. Maternal urinary proteome detection was earlier than the developmental time point of tissue sections observed by microscopy.

## Figures and Tables

**Figure 1 biomolecules-13-00034-f001:**
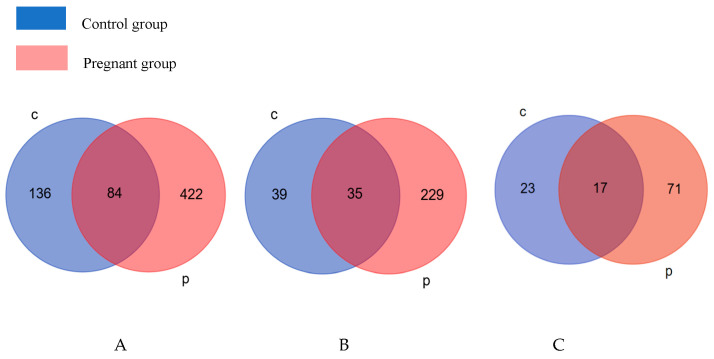
Differential proteins at all time points in the control and pregnant groups (**A**); biological processes at all time points in the two groups (**B**); IPA classical biological pathways at all time points in both groups (**C**). C: control group; P:pregnant group.

**Figure 2 biomolecules-13-00034-f002:**
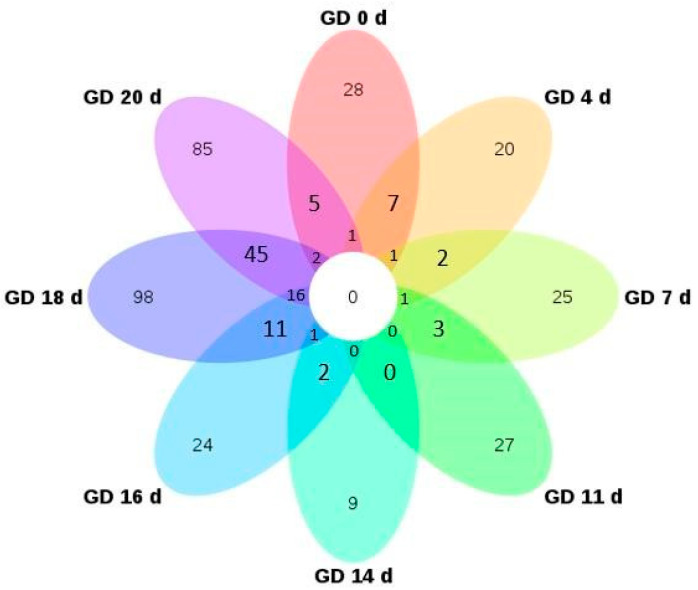
Venn diagram of differential proteins in the pregnant group.

**Figure 3 biomolecules-13-00034-f003:**
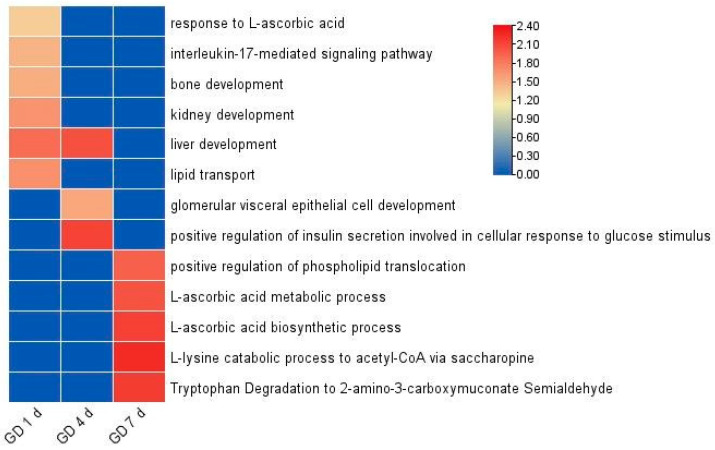
Biological pathways of differential proteins on GD 1 d, GD 4 d, and GD 7 d by analyzed GO.

**Figure 4 biomolecules-13-00034-f004:**
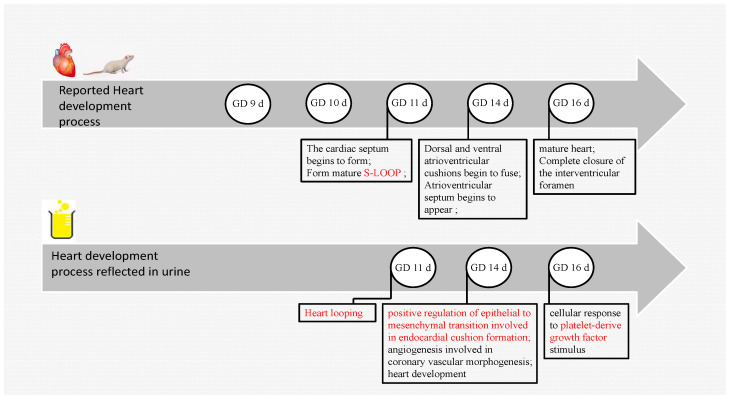
Fetal rat heart development. The above is the reported heart development process. The following are cardiac developmental processes enriched by changing urinary proteomes.

**Figure 5 biomolecules-13-00034-f005:**
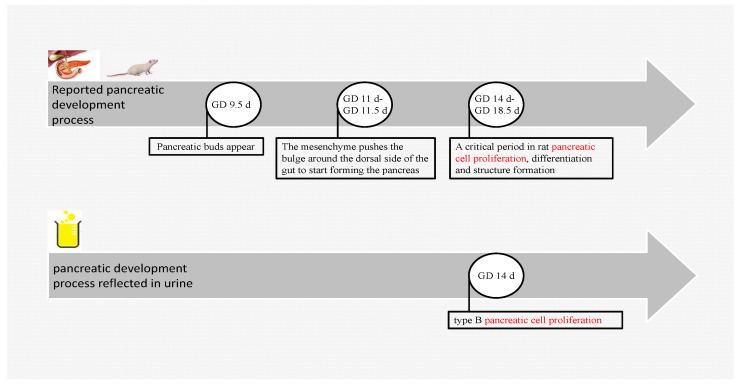
Fetal rat pancreas development. The above is the reported pancreatic development process. The following are pancreatic developmental processes enriched by changing urinary proteomes.

**Figure 6 biomolecules-13-00034-f006:**
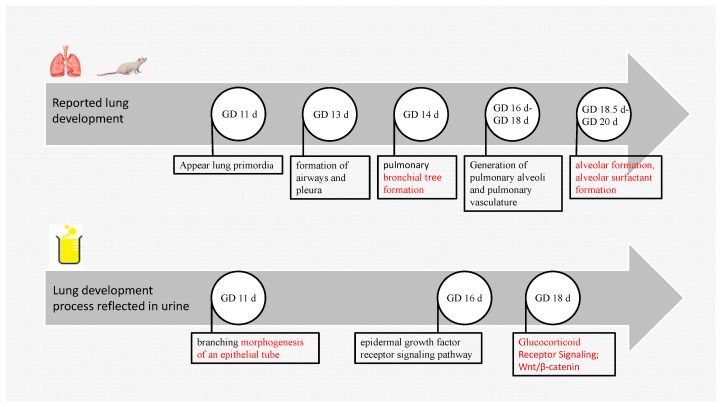
Fetal rat lung development. The above is the reported lung development process. The following are lung developmental processes enriched by changing urinary proteomes.

**Figure 7 biomolecules-13-00034-f007:**
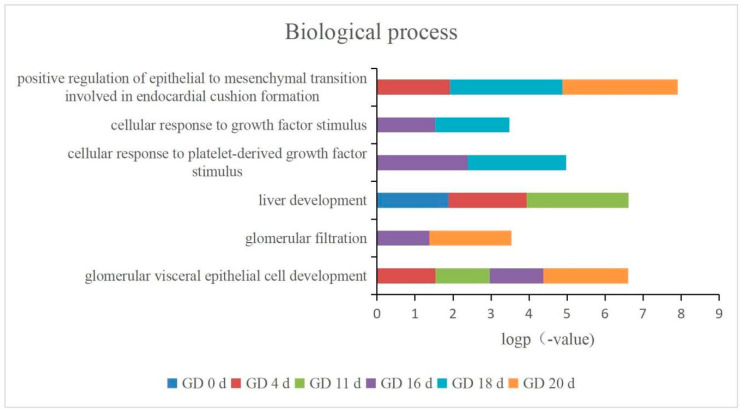
Changes in pathways related to organ development over time.

**Figure 8 biomolecules-13-00034-f008:**
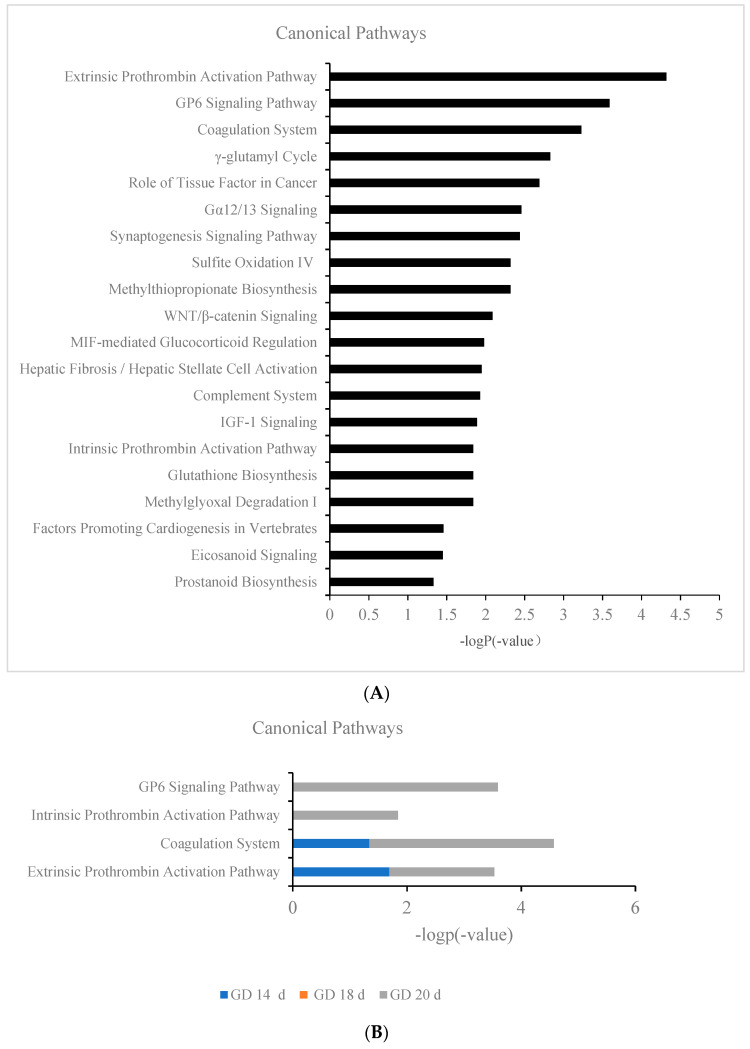
(**A**) Biological processes by IPA (*p* value < 0.05) at GD 20 d. (**B**) Changes in coagulation-related pathways at GD 14 d, GD 18 d, and GD 20 d.

**Figure 9 biomolecules-13-00034-f009:**
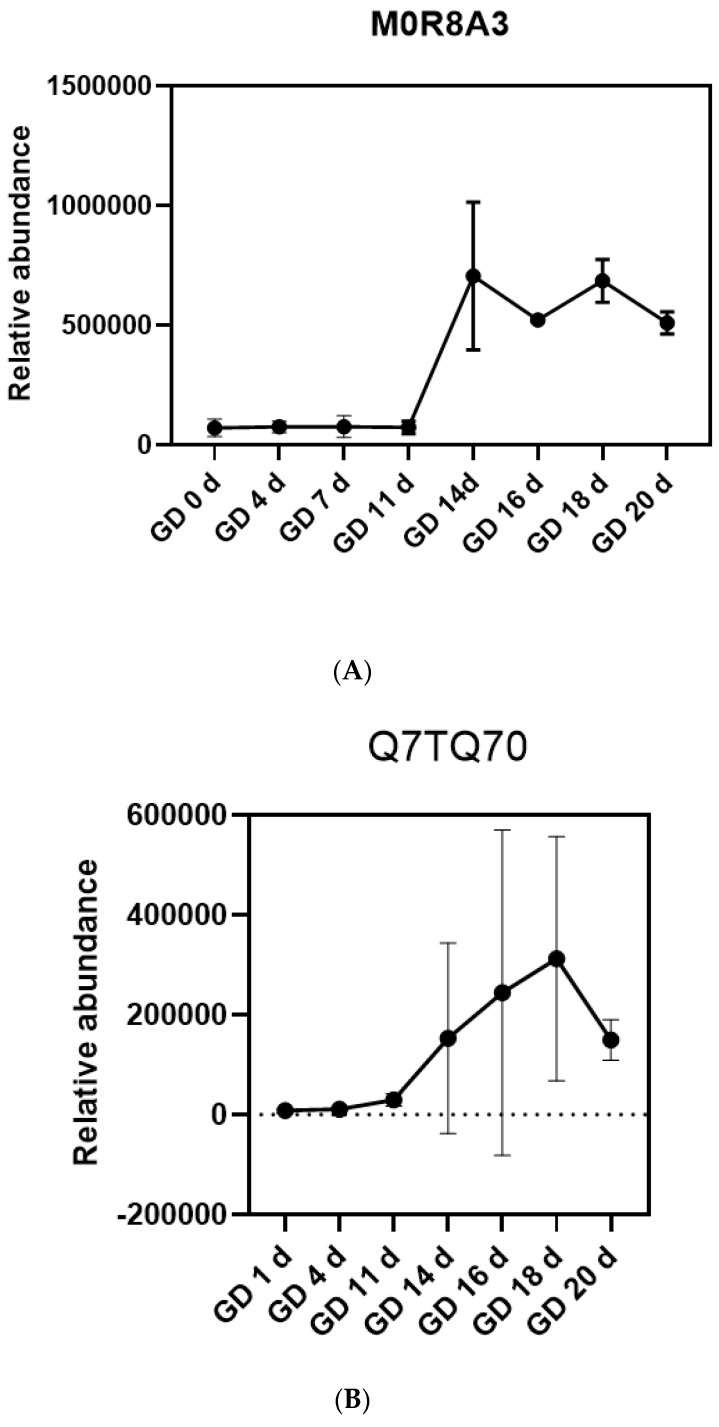
The urinary proteome exhibited coagulation-related changes before labor. (**A**) The line plot for the relative abundance of coagulation-related protein MOR8A3 during pregnancy. (**B**) The line plot for the relative abundance of coagulation-related protein Q7TQ70 during pregnancy. Each group (time point) was involved in three experimental replicates. Relative protein expression levels at each time point were assigned the mean ± SD value. (**C**) Coagulation-related proteins MOR8A3, Q7TQ70, and Q8K3U6 on GD 14 d, GD 16 d, and GD 20 d.

**Table 1 biomolecules-13-00034-t001:** Urine proteome quantitative results.

Batch		Time Point	Number of Identified Proteins	Number of Proteins after Data Preprocessing
First batch	Pregnant group	GD 1 d, GD 14, GD 16 d, GD 18 d, GD 20 d	1703 ± 277	1199
Second batch	Pregnant group	GD 1 d, GD 4 d, GD 7 d, GD 9 d, GD 11 d	1498.6 ± 249	1289
	Control group	0 d, 4 d, 7 d, 9 d, 11 d, 14 d, 16 d, 18 d		

Number of identified proteins are given as mean ± SD.

## Data Availability

The datasets presented in this study can be found in online repositories: http://www.proteomexchange.org/. The database can be found since publication.

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
