# Peer review of "Urinary Proteome Changes during Pregnancy in Rats"

_biomolecules, 2022, doi:10.3390/biom13010034_

Round 1

Reviewer 1 Report

The authors Tan and Gao presented a work describing the urine proteome changes during pregnancy in rats. 

Major Comments:

Introduction Section

1. The introduction is little confusing. 18 references were mentioned in the introduction, where most of them are studying humans. As  there are several other studies for urinary proteomics in humans via analytical methods (e.g., 10.21037/atm-21-3497,10.1186/1471-2164-14-777, 10.3390/metabo11110723) are available, the aim of the presented study is not clear. What is known from pregnant animals/rats? Was the intention, to develop a new analytical method? 

2. Lines 55-64 and Figure 1 describes study design. Study design should be moved to method section.

Method Section

3. Please specify time point and method for urine collection from pregnant and control group in the method section.

4. Where all samples prepared for measuring directly upon collection or where they stored.  Please specify.

5. Did Figure 10A,B show protein expression? A description in the method section for sample type, preparation and detection method is missing. 

Results section

6. Why samples collection was done in two batches from pregnant rats?

7. Suppl. Table 1: data from 0 d in the control group is missing.

8. Suppl. Table 4: data for IPA canonical pathways day 16 is missing.

9. Figure 10A and 10B: Please state animal number per time point in figure legend.

10. Line 313: Please give the references for the mentioned "previous studies".

11. Line 314: Please give the references for the mentioned "previous studies".

Minor Comments

a) Please check text and figures for typos.

b) Please introduce abbreviations when used for the first time and use them consequently afterwards.

c) Figure 2: color code and letter code for control and pregnant group is little confusing. Letter were used in as capital letter in (C) but not in (A)/(B).

d) Section 3 is "Results and Discussion", point 4 "Discussion". Please use another heading for section 4. Discussion part is confusing. Please specify the main novel finding of the current study.

e) Line 295: pregnant mice or rats?

Reviewer 2 Report

Urinary proteome changes during pregnancy in rats

The aim of the paper is to explore whether the rat urinary proteome could reflect maternal changes and the process of embryonic growth and development.

The paper is interesting and in general clearly written. The analytical protocols are appropriate. However, the experimental design, the experimental animals are incomplete and require additional explanations. Moreover, the statistical analysis section is unclear and needs revision and improvement.

Major revisions

How was the urine sampled? Free catch? Cystocentesis? This must be indicated.

When was sampled the urine? Morning urine?

Lines 76-80: “The first batch of urine samples was collected from pregnant rats (n=10): GD 1d, GD 14 d, GD 16 d, GD 18 d and GD 20 d. The second batch of urine samples was collected from pregnant rats (n=5) and control rats (n=5): pregnant group: GD 1 d, GD 4 d, GD 7 d, GD 11 d; control group 1 d, 4 d, 7 d, 11 d, 14 d, 16 d, 18 d, 20 d.” It is unclear why the pregnant rats were divided into two nonhomogeneous groups (10 and 5). First group seems used for samplings related to long-term embryonic development and the other for short-term development.

It is well known that the urinary proteome is subjected to wide intraindividual and interindividual variations, why two separate batches on different rats?

Were the non pregnant control rats of the same age as those pregnant? This important information is lacking.

It is also unclear why only 5 control rats were used.

These issues must be addressed.

Moreover, the flow chart in Figure 1 does not reflect what reported at lines 75-80.

Line 83: Urine (4-6mL): only une sample was analysed for each sampling time? Was this sample obtained after pooling the urine of different rats? These information are lacking.

Was the urine analysed immediately after the sampling?  

Statistical analysis: this section is lacking and in need of deepening and supplementation: The authors say they used ANOVA for comparison “One-way ANOVA was used for the comparison of data at different time points. The proteome at each time point in the pregnant and control groups was compared with the 0 d (GD 1 d) proteome.”

Was a test of normality done? Was a test of homoscedasticity done? Without meeting these requirements, in the reviewer’s opinio, ANOVA for repeated measures cannot be applied. If data are not nornally distributed, other non-parametric tests, as Friedman test can be used.

Line 137: t-test is mentioned: for which comparison was it used?

Line 138: “All results are presented as the mean ± standard deviation (SD)”: mean of which and how many values?

Was statistical analysis performed to analyse the differences between pregnant and non pregnant rats for each timepoint?

Table 1: “Mean ± SD of the identified proteins” is it an average between the proteins identified in the samplings at different times of pregnancy? Why is there no standard deviation after processing? No identified proteins in the control group?

Was statistical analysis performed on data reported in figures 10 A and B?

lines 304-305: “P value was more significant over time” P value is significant at 0.05 or 0.01, “more significant” in not informative.

Lines 307-320 could be moved to the conclusion section.

Minor revisions

- lines 50-52 are referring to urinary proteome variations in some diseases. It would be more appropriate to cite articles regarding urinary proteome changes during pregnancy.

- line 67: female Wistar rats: replace “female” with “control”

- line 149: the number of urine proteins

- line 153: “credible proteins” do you mean unambigously identified?

- line 166: Venn diagram

-line 198: Fig 4. instead fig 5.

- line 209: Changes in urinary proteome

-line 295: pregnant rats?

- lines 299-304; please check punctuation

- figure 10°: data are reported as mean±SD? Please add this information to the caption of the figure

- line 331: “4. Discussion”, change to “4. Conclusion”

- line 329: an effective tool?

- Figures 5-7: In the second arrow, align the circle containing GD with those shown in the first arrow.

Round 2

Reviewer 2 Report

The manuscript has been modified. However, some parts remain that need to be modified.

-The introduction has been shortened; however, in the reviewer opinion, this revised version does not provide sufficient background on the specific topic and still  citations which are not related to pregnancy remain. In addition not enough emphasis has been given to the aim of the research: "So, we intended to explore if urine can reflect pregnancy and fetal development processes".

-lines 99-102: in the reviewer opinion this sentence should be moved under 2.5 Data Processing.

-table 1: heading "mean ± SD of the identified proteins" should be "number of identified proteins"?

-"Fold change and P value of the procoagulant related proteins became larger over time": the sentence needs to be modified, because P value cannot become larger over time.

- Figures 4-6:the position of circles look just like in the first unrevised version of the manuscript.
